Subject Category:
Biology (whole organism)

Subject Areas:
evolution/cognition/behaviour

Keywords:
cultural evolution, language change, kinship systems, language use

Author for correspondence:
Péter Rácz
e-mail: raczp@ceu.edu

# Usage frequency and lexical class determine the evolution of kinship terms in Indo-European

Péter Rácz[1,2], Sam Passmore[2], Catherine Sheard[3,4] and Fiona M. Jordan[2]

[1]Cognitive Development Center, Central European University, Budapest 1051, Hungary
[2]Department of Anthropology and Archaeology, University of Bristol, Bristol BS8 1UU, UK
[3]School of Biology, University of St Andrews, St Andrews KY16 9TJ, UK
[4]School of Earth Sciences, University of Bristol, Bristol BS8 1TQ, UK

PR, 0000-0001-7896-4801; SP, 0000-0002-5302-356X;
CS, 0000-0002-8259-1275; FMJ, 0000-0002-9953-8884

Languages do not replace their vocabularies at an even rate: words endure longer if they are used more frequently. This effect, which has parallels in evolutionary biology, has been demonstrated for the core vocabulary, a set of common, unrelated meanings. The extent to which it replicates in closed lexical classes remains to be seen, and may indicate how general this effect is in language change. Here, we use phylogenetic comparative methods to investigate the history of 10 kinship categories, a type of closed lexical class of content words, across 47 Indo-European languages. We find that their rate of replacement is correlated with their usage frequency, and this relationship is stronger than in the case of the core vocabulary, even though the envelope of variation is comparable across the two cases. We also find that the residual variation in the rate of replacement of kinship terms is related to genealogical distance of referent to kin. We argue that this relationship is the result of social changes and corresponding shifts in the entire semantic class of kinship terms, shifts typically not present in the core vocabulary. Thus, an understanding of the scope and limits of social change is needed to understand changes in kinship systems, and broader context is necessary to model cultural evolution in particular and the process of system change in general.

# 1. Background

Languages change over time: sounds, syntax and vocabulary are all in varying states of flux due to the shifting social equilibrium

in which language users interact. Innovations arise in populations, and variants undergo filtering processes that determine their rise or fall in the long run. Older views of language evolution saw these processes as a force of nature, severed from and unaffected by the dynamics of the speaker community [1], but those perspectives have been largely replaced by work in variationist sociolinguistics [2] and usage-based linguistics [3] which relate language change to social and cognitive biases in the individual and patterns in the community.

Sociolinguistic research that focuses on the individual-level aspects of how language is learned and transmitted in the community [4,5] is mirrored by a predominantly micro-evolutionary [6,7] approach to human social learning in general [8,9]. This approach has a primary focus on modelling the dynamics of cultural variation within populations by considering the effects of biased social learning on the fate of cultural features, either due to their inherent properties (content), or aspects of the transmission process (context) [8].

The relationship between micro- and macro-evolutionary dynamics of cultural evolution is still not well understood: for example, what generalities can be made in how changes in speaker behaviour ramify up to differences between languages? Historical linguistics has increasingly drawn on the theory and methods of evolutionary biology to provide macro-evolutionary frameworks for understanding and analysing patterns of language relatedness, diversification and change. Phylogenetic inference of large-scale language family relationships is increasingly commonplace [10] due to the development of quantitative models appropriate for lexical replacement [11]; in turn, analysts are able to estimate dates of diversification and age of origin of related languages [12–14] without assuming that language changes at constant rates.

Phylogenetic methods have also demonstrated that the rate of replacement in the core vocabulary of a language correlates with its usage frequency in large-scale linguistic corpora [15]. The core vocabulary is a predefined set of frequent words such as 'heart', 'walk', 'bone' and 'good' [16]. For example, while German retains the Proto-Germanic form for 'chair', 'Stuhl', English has replaced it with the Latin form, probably as a result of rising Franco-Norman influence in the language's history. However, both German and English retain the Proto-Germanic form for 'man', which is a much more frequently used term than 'chair' and a part of the core vocabulary. This so far appears to be a robust correlation, as words for more frequently used meanings are replaced at a slower rate in core vocabulary [15] and in numerals [17], prompting the claim that frequency of usage could provide a cross-linguistic general mechanism for the rate differences in lexical replacement [15]. An analogous process exists in biological evolution, where highly conserved (slowly evolving) regions of genetic material are generally considered to have increased functional utility in biological processes compared with faster-evolving regions [18].

In language, the mechanisms responsible for the shielding effects of frequency of use remain unclear. While language change ultimately occurs at the individual level, it is very unlikely that a word's frequency of occurrence in a particular language's corpora maps directly onto its rate of replacement for several reasons. The two factors operate on different time-scales, frequency of occurrence is at best an indirect measure of the strength of a word's lexical representation [19], and word frequencies are themselves correlated within a language corpus [20,21]. This is an especially pertinent aspect of words in closed classes of content words (such as number or colour systems), where the universe of potential meanings and referents is relatively restricted, because a change in one part of the system may have knock-on effects that ramify throughout [22,23].

In most descriptions, the terminology of open/closed and content/function are used interchangeably for word classes. Here, we step away from this and use 'closed semantic class' to refer to a set of closely related words where the head words can be listed exhaustively. By contrast, an open semantic class has loose (if any) connections between its members and can theoretically be of any size.

Human kinship is an extensively studied example of a closed semantic class; specifically, the terminological (lexical) systems used to refer to relatives [24]. Kinship terminologies are cultural as well as linguistic phenomena, reflecting a society's wider social norms governing relatedness, marriage, inheritance and family organization. Despite a breadth of diversity in social norms, kinship terminologies show far less variation than is combinatorially possible. This lack of diversity is at least partially constrained by our biological inheritance as social primates [25,26], but equally by cognition [27] and strong patterns of cultural inheritance [28]. Kinship terminologies provide us with a platform to explore the interaction of biological and cultural domains, and their effect on language use and change [29].

Kinship words together constitute systems in the sense that they have a culturally circumscribed range of referents (individuals are or are not kin; if so, are different kinds of kin). Importantly, they pattern together. For example, Swedish distinguishes *father's father* and *mother's father*, and Polish distinguishes *sister's daughter* and *brother's daughter*, while English makes no such contrasts: those

**Table 1.** Kinship terms for father's father (FF), mother's father (MF), sister's daughter (ZD) and brother's daughter (BD) in Swedish, Polish and English.

| | Swedish | Polish | English |
|---|---|---|---|
| FF | *farfar* | *dziadek* | *grandfather* |
| MF | *morfar* | *dziadek* | *grandfather* |
| ZD | *syster-dotter* | *siostrzenica* | *niece* |
| BD | *brors-dotter* | *bratanica* | *niece* |

relatives are, respectively, grandfathers and nieces. When a kinship system changes, words are replaced in clusters: a shift from a Swedish-type system to an English-type system could erase *both* the term for *father's father* and for *mother's father* (table 1).

This means that a kinship term might be replaced in itself, but this change will tend to propagate over the range of related terms. The rate of replacement for a given term is determined by shifts in the entire semantic class rather than by the frequency of the term itself.

The replacement of the semantic class, while ultimately contingent on usage frequency (words no longer said will disappear), is subject to complex social and cultural pressures [28,30]. Typological work suggests that kinship changes on the level of entire semantic classes rather than individual words, but the patterns of change in form and meaning in kinship terminologies are far from systematically surveyed (see [31]). The aim of this work is to address this issue in kinship research and thereby to shed light on how frequency effects in language change are mediated by semantic classes.

## 2. Questions

The rate of replacement of the Indo-European core vocabulary, a set of frequent words like 'heart', 'walk', 'bone' and 'good' that constitute an open semantic class, correlates with their usage frequency [15]. In this paper, we use phylogenetic comparative methods to analyse kinship words in Indo-European and address the following questions. (i) Do kinship words show the same effect of usage frequency on the rate of replacement as core vocabulary words? That is, can frequency effects on the core vocabulary be replicated with a closed semantic class? (ii) Given that kinship words are argued to shift together as a class rather than individually and incrementally, does this result in a difference between the effect of usage frequency for core vocabulary words and for kinship terms?

## 3. Methods

To address these questions, we selected a set of widely shared kin categories and collected the terms used for those kin-types in 47 Indo-European languages. We then sampled the word usage frequencies of each kin term in a range of corpora from a subset of those languages (mostly from Europe) and determined the rate of replacement for each term to compare replacement rate and usage frequency, following earlier work on the core vocabulary [15].

Data were collected for ten basic consanguineal kin term categories. The categories cover parents, children, siblings, aunts/uncles and cousins, while the languages represent five main branches of Indo-European.

We then generated cognate classes for the specific forms across the language sample and paired these with a phylogeny of Indo-European languages to determine the rates of replacement for the individual terms. This allowed us to estimate the rate of replacement for each basic kin term category. We followed similar work on the core vocabulary that used the 200-item Swadesh list [16], a compilation of fundamental lexicalized concepts used in comparative linguistics. This practice works at a greater resolution and consequently entails a higher level of specificity than specifying cognate historical relationships. For instance, we consider German 'bruder' and French 'frére' as members of two different cognate classes given both their formal and etymological distance.

Figure 1 shows the classes in colour for the words expressing mother's sister (MZ) and brother (B) in our dataset across languages in the Indo-European family tree. The types of terms used for MZ are more diverse than those used for B; taking into consideration the genealogical history of these terms, we can

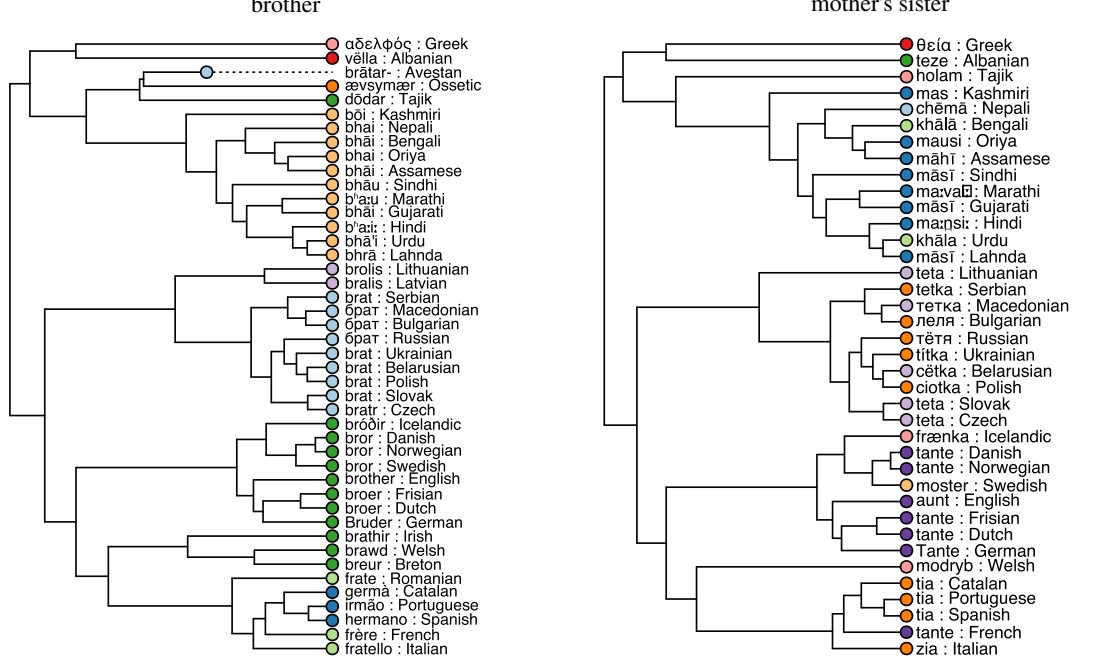

**Figure 1.** The Indo-European phylogeny and cognate classes (indicated by colour) derived by LingPy and checked by linguists for the terms brother and mother's sister (aunt).

estimate that this difference in diversity stems from the fact that B terms are replaced slower than MZ terms on a historical time-scale.

The rate of replacement estimates were then compared with frequency of use across corpora.

## 3.1. Kinship data

As part of a larger database project on kinship terminology, we collected kin terms from 47 languages for the following relatives: brother (B), daughter (D), father (F), mother (M), maternal uncle (MB), maternal aunt (MZ), maternal aunt's daughter (MZD), maternal aunt's son (MZS), son (S) and sister (Z). Most Indo-European languages do not distinguish lineal relationships with separate terms, such as maternal versus paternal uncle (e.g. MB versus FB), and if they do such terms tend to be rare in contemporary kinship terminologies. Therefore, we used terms for maternal relatives as representative. We have not included words for spouses (e.g. husband or wife), because these are often the same word as 'man' or 'woman'. Frequency data were collected from 34 corpora in 21 languages in three corpus types: spoken, written and web-crawled. The sources for the terms and the corpora are listed in the electronic supplementary material.

## 3.2. Cognate data

We generated cognate classes by using the Indo-European Etymological Dictionary [32], LingPy [33] and expert judgement, inviting volunteers on Linguist List to review the cognate classes. All terms were first automatically transcribed into the Speech Assessment Methods Phonetic Alphabet (SAMPA) through LingPy's `uni2sampa` function. Cognates were automatically allocated using LingPy's cluster function with an edit-distance algorithm and 0.4 threshold. These categories were then reviewed by expert volunteers. Automatic decisions and expert corrections are available in the electronic supplementary material. To compare kinship terms with basic vocabulary, we obtained measures of frequency and rates of replacement for meanings on the Swadesh list. Measures for frequency of use in Swadesh terms for English, Spanish, Russian and Greek, and the rates of replacement for Indo-European languages for Swadesh terms were obtained from the electronic supplementary material of [15], which we follow in our methods. Measures for frequency of use in Swadesh terms for Portuguese, French, Czech, Polish and German come from the electronic supplementary material of [34].

## 3.3. Phylogeny

To estimate rates of replacement and to account for phylogenetic uncertainty, we used 1000 phylogenies from the most recent Bayesian posterior of Indo-European phylogenies [13]. Trees in the sample are rooted. Branch lengths are given in years and derived from statistical and historical calibration, which dates Proto-Indo-European to approximately 8700 years ago. Trees initially have 111 taxa, and these were pruned for each kin term to match the available data.

## 3.4. Rates of replacement

Following the methods in [35], we used BayesTraits v. 3.0.1 to implement a phylogenetic Bayesian MCMC approach to estimate the instantaneous global rate of replacement for each kin term through Q-matrix normalization. Probabilities of frequency were scaled to represent the empirical frequencies. We used a stepping-stone sampler, using 100 stones for 1000 iterations each. MCMC chains run for a total of 10 010 000 iterations, with a burn-in of 10 000, sampling every 1000. This leaves a posterior sample of 10 000 iterations (approx. 10 iterations per tree). To make the rates comparable to previous work, we scaled instantaneous rates to change per 10 000 years. For each kin term, we ran MCMC chains three times, to ensure convergence. The averaged results of each run, as well as Gelman–Rubin diagnostic tests of MCMC-chain convergence are available in the electronic supplementary material [36].

## 3.5. Usage frequency and rates of replacement

We used the lme4 package in R [37,38] to fit a mixed-effects generalized logistic regression model to the combined data (basic vocabulary and kinship terms). The outcome variable is the mean rate of replacement estimated for a word meaning. The predictor variables are aggregated centralized log frequency per million (clfpm) per word per language and word type (Swadesh term or kin term). Swadesh terms that are kin terms (mother, father, husband and wife) were excluded (for details, see the electronic supplementary material).

The random effect structure of the reported model was selected using model comparison based on goodness-of-fit tests and the Akaike information criterion.

# 4. Results

Our main questions were (i) whether kinship words show an effect of usage frequency on the rate of replacement and (ii) whether this effect is different for kinship words and words in the core vocabulary. In order to answer these questions, we modelled the rate of replacement of ten kin terms in Indo-European and, in turn, compared (i) the rates of replacement to the frequency of use in language corpora and (ii) our kin term data with available data on core vocabulary terms.

We found a negative correlation between how often a kin term is used in the languages in our sample and its estimated rate of replacement (est = −0.58, s.d. = 0.07, t = −8.72). Genealogically close terms like 'mother', 'sister' or 'son' are used more frequently and change slower. By contrast, more distant terms like 'mother's sister' (English: 'aunt', French: 'tante', etc.) are less frequent and change faster. In this respect, kin terms behave like core vocabulary: more frequent terms change slower in linguistic evolutionary history.

Figure 2 shows the log frequency distribution of our terms across the languages in the sample (x-axis). The terms themselves are sorted according to their estimated rate of replacement. Mother's (or father's) sister (MZ), which has the highest estimated rate of replacement, is on top. Brother (B), which has the lowest rate, is at the bottom. While the relationship is not uniform, we see that, overall, terms that are used more frequently are also replaced slower across languages.

In addition, this relationship between frequency of use and replacement is *stronger* for kin terms than for general vocabulary terms (est = 0.18, s.d. = 0.09, t = 2.02). Kin terms used more frequently change slower, while those used less frequently change faster than terms in the core vocabulary. This can be seen in figure 3, which shows the mean log frequencies (x-axis) of kin terms (red labels) and Swadesh terms (grey points) across their mean rate of replacement (y-axis). The regression line expressing the strength of the correlation is steeper for kin terms than for Swadesh terms. The envelope of variation for kin terms and core vocabulary is similar. Since kin term frequencies were sampled from a range of

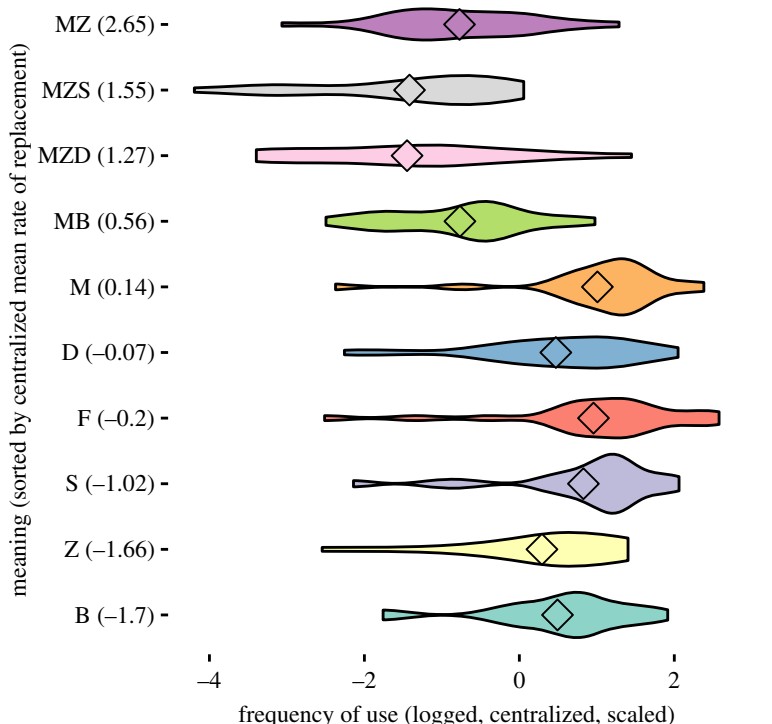

**Figure 2.** Usage frequency across languages for kin terms in the data (sorted by the term's mean rate of replacement).

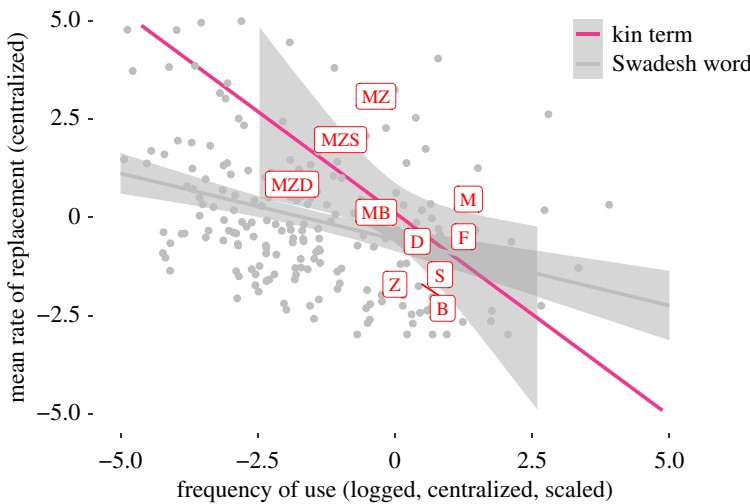

**Figure 3.** The relationship between the mean rate of replacement and the mean usage frequency for Swadesh words (grey) and kin terms (red), with a simple linear model.

corpora (encompassing different registers, *viz.* written, spoken and web-based language), this remains a robust result.

In sum, kin terms resemble the general vocabulary in that respective words used more frequently are more resistant to being replaced. This effect is more drastic for kin terms, which are even less likely to be replaced when they are frequent and even more likely when they are rare, in comparison with the core vocabulary.

# 5. Discussion

The idea that closed lexical classes vary in their rate of replacement has received attention [35]. Yet, to our knowledge, ours is the first study that systematically investigates the behaviour of a closed semantic class in comparison with an open class (the items of the core vocabulary) using corpus data and phylogenetic methods.

Our results indicate that a closed semantic class is not simply more or less volatile than an open semantic class in this regard. The systematic difference in the strength of the interaction between frequency and rate of replacement potentially stems from the nature of change in kinship systems. Changes may be concerted and systematic over a category of relatives, rather than independently replacing words one by one. This, in turn, is because kinship systems incorporate conceptual reflections of real-life social structures of family and kinship, and they are affected by changes in social conditions, societal norms and inheritance. For example, the Latin kinship system distinguished paternal uncles (FB, *patruus*) from maternal (MB, *avunculus*); towards the end of the Roman republic; however, *avunculus* was being used bilaterally, with similar changes occurring in parent's sister and cousin terms [24]. These linguistic changes are argued to reflect the rise of a religious belief in monogamy, nuclear-family focused peasantry and restrictions on degrees of allowed marriage.

The peculiar evolution of kinship terms is therefore not an idiosyncratic quality, but rather one to be explained through the details of the process. When we look at Indo-European kinship, we see that often-invoked filial (brother, sister) and spousal (wife, husband) terms change rarely.

More distant consanguineal terms are more prone to shifts, including levelling scenarios where terms may sweep over previously differentiated relatives. One example is the rise of 'cousin' and its minor variants (German 'Cousin/e', French 'cousin/e', Polish 'kuzyn/ka') to cover all parents' siblings children across Europe (visible in our data); cross/parallel distinctions were previously attested in different branches of Indo-European [39].

The more abrupt and wider changes in semantic systems neatly account for the observed behaviour of kinship terms in our sample. Core terms, like 'father' and 'daughter' are replaced disproportionately slower, because semantic shifts between kinship systems are less likely to affect these terms. By contrast, more distant terms, like 'aunt' and 'cousin', are replaced in such shifts more often. While we need to speculate at this point, it is ultimately also likely that age of learning ('mother' learned earlier than 'aunt') and frequency of use play a role here [40].

One caution needs to be raised due to the atypicality of the Indo-European language family. It is no accident that globally, language corpora and phylogenetic information are most readily available for Indo-European languages. These are mostly spoken in industrialized nation-states with a long history of language documentation and scholarly attention. Recent research [41] argues that shifts in European kinship, spread via the Christian Church in the Middle Ages, promulgated democracy and industrialization in Europe. If this is the case, we may be able to study language change in these populations indirectly because of kinship dynamics; however, the effect may be on the generality of these results rather than a bias in outcome.

More broadly, our sample of Indo-European languages over-represents WEIRD (Western, educated, industrialized, rich, democratic) societies [42] as well as so-called Eskimo bilateral kinship systems [43] and thus limits generality. Across Europe we also find a strong correlation between genealogical relatedness and geographic distance (siblings live closest, grandparents or grandchildren live a little further, and first cousins live furthest; [44]). This is unlikely to be a global cross-cultural pattern, but exemplifies the relationship between social organization and kinship terminologies.

Recent work [45] has considered the relationship between kinship terms and fitness interdependence, which is the degree to which organisms influence one another's evolutionary fitness, and goes beyond the standard coefficients of relatedness. This work predicts that 'Eskimo' systems in particular will be found where there are strong differentials in the fitness interdependence of 'nuclear' family members versus other relatives, which is in line with the exaggerated relationship we find in the lexical frequency–change relationship. 'Eskimo' systems, in particular, tend to have a smaller kin term inventory than systems that make more distinctions, for example, 'Sudanese'. Inventory size might mediate the ability to detect any frequency–change relationship, and this could be tested if corpora were available for a wider range of languages. We encourage investigations of usage frequency, lexical classes and word evolutionary rates in other language families.

On a more general note, historical linguists can find comfort in our results, which provide additional support to the use of the core vocabulary to establish a correlation between frequency of use and rate of replacement. However, these results also indicate that closed semantic classes behave differently and so across-the-board treatments of language change result in a loss of valuable resolution. The distinctness of kinship terms probably generalizes to other closed classes. This invites further cross-linguistic research on word frequency and language change in other classes, such as colours or numerals [23,46].

In this paper, we used a phylogenetic analysis of kinship words in Indo-European languages to show that for kinship terms, cultural macro-evolutionary patterns are partially mediated through exaggerated frequency effects. The creative analogy of parallels between biological and cultural evolution is

heuristically valuable and opens up a diverse hypothesis space for unique aspects of human culture such as kinship. The replacement rate of genes, like that of words, is at least partially dependent on their function [18]. Anthropologists and linguists hold rich resources of empirical data that can be used to elaborate the operationalization of 'cultural transcription'. Here, we have drawn on different genres of language corpora for usage frequency: web corpora, in particular, provide access to a wide range of naturalistic language at much less cost than curated national corpora. Furthermore, classes of traits that evolve in concert, and particular phenotypes (or traits) that are more or less likely to arise due to properties of the system as a whole, are phenomena that occur across evolving systems: biological processes of evolvability [47] and developmental bias [48,49] could provide useful analogies for future research.

Our results that changes within semantic classes are correlated at the macro-evolutionary scale and thus that cultural context can constrain language change, emphasize the need for careful application of the appropriate evolutionary methods to the study of cultural evolution.

Data accessibility. All our data and code are available in the extended electronic supplementary material hosted at doi:10.5281/zenodo.3453517.

Authors' contributions. P.R., S.P., C.S. and F.M.J. designed research; P.R. collected corpus data; S.P. and F.M.J. collected kinship terms; S.P. and C.S. performed phylogenetic analyses; P.R. performed regression analysis; P.R. wrote the first draft of the manuscript and S.P. the first draft of the electronic supplementary material, with all authors contributing writing and editing. P.R. and S.P. contributed equally to this work.

Competing interests. We declare we have no competing interests.

Funding. This research was funded by the European Research Council's Horizon 2020 programme under Starting Grant no. 639291 VARIKIN to F.M.J.

Acknowledgements. We thank Johann-Mattis List for help with LingPy, Seán Roberts for his cyfarwyddyd, and Hannah Booth, Mate Kapovic, Aaditya Kulkarni, Nikolche Mickoski, Rajalaxmi Pradhan, Yilana Rodriguez, Jason Rogers, Geoff Sampson and Ollie Sayeed who volunteered to check our cognate classes. All faults remain ours.

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
