## [Reviewer comments · Royal Society Open Science]

Review History

RSOS-191385.R0 (Original submission)

Review form: Reviewer 1 (Andreea Calude)

Is the manuscript scientifically sound in its present form?

Yes

Are the interpretations and conclusions justified by the results?

Yes

Is the language acceptable?

Yes

Do you have any ethical concerns with this paper?

No

Have you any concerns about statistical analyses in this paper?

No

Recommendation?

Accept with minor revision (please list in comments)

Comments to the Author(s)

This paper concerns the evolution of kinship words in Indo-European languages. The two main questions asked are:

- (1) is the rate of replacement of kinship words in IE languages mediated by frequency of use of these words?
- (2) if so, is the rate comparable with other core vocabulary or similar to it?

The findings suggest that (perhaps unsurprisingly) central kinship words behave similarly to other core vocabulary words, in that rates of replacement are higher for infrequent words and lower for frequent words. The most newsworthy finding is - to my mind - the answer to the second question, namely that the effect is stronger for kinship words than for other core vocabulary words. In other words, if we believe the frequently-used-leads-to-slowly-replaced paradigm, then kinship words might be considered the more prototypical examples of such trends.

The only real quibble I had was with regard to the use of the term "closed lexical class". Generally speaking, "closed classes" refer to syntactic categories (nouns, verbs, adjective, etc.), whereas this paper is dealing with a semantic category (i.e., kinship). I am personally not familiar with the terms "closed"/"open" being used in relation to semantic classes (as used here).

Further to this, I am not sure which "open class" the authors have in mind with regards to the comparison with kinship words - picked up in the discussion (this is the assertion on page 10-11, lines 275-278). I am not necessarily disagreeing with the assertion made there, but that first point in the discussion had me rather confused.

I was also wondering if the authors may have considered comparing the strength of the relationship between frequency of use and replacement words of kinship words with the relationship of frequency and replacement rates of number words. Number words are similarly a "system" semantic group - in that number words form a system together, in a way that kinship words may be said to form a system together. There is an older paper by Dehaene, S., & Mehler, J. (1992) looking at frequency of use of various number words in a bunch of IE languages which is interesting in this regard (see reference details at the end of this review).

As a general comment, I think the discussion could be developed further. For instance, this discussion of kinship words poses a few questions for future research: (1) in light of these results, should core vocabulary lists include more kinship terms, (2) are these results about kinship words specifically, or about groups of concepts which form such a "system" together (like kinship, numbers, maybe pronouns)? (3) does this trend uncovered for kinship words relate to kinship, or might it be a correlate of the fact that kinship terms are generally monosemous (rather than polysemous)?

Overall, the paper makes a worthy contribution to the development of kinship systems in IE languages and to language change in general and as such, I feel it should be published. The organisation and language of the paper are both of very high standard.

One small typo I noticed, page 12, line 334: a space is needed after the colon.

Reference:

Dehaene, S., & Mehler, J. (1992). Cross-linguistic regularities in the frequency of number words. *Cognition*, 43(1), 1-29.

Review form: Reviewer 2

Is the manuscript scientifically sound in its present form?

Yes

Are the interpretations and conclusions justified by the results?

Yes

Is the language acceptable?

Yes

Do you have any ethical concerns with this paper?

No

Have you any concerns about statistical analyses in this paper?

Yes

Recommendation?

Accept with minor revision (please list in comments)

Comments to the Author(s)

The authors repeated a previous study on the correlation between word usage frequency and its rate of replacement along phylogeny on a closed lexical class. This is the first study looking at this problem on a closed lexical class, so the study definitely provides new insights into language evolution. I don't have the expertise to comment on the data. In terms of the statistical analyses, the study uses basically the same approach as the previous study, so the analyses seem to be valid. However, there is few description on the linear model. I have to go to the supps for details. Yet, even in the supps, the description is not clear, only two tables listing the fixed effect, without any explanation on what is control model 1 and control model 1b. In the main text, it says that the random effect structure was selected using model comparison. But there is no corresponding result and no description on the tested random effect structure. So I recommend the authors add more description on the linear model.

Decision letter (RSOS-191385.R0)

08-Sep-2019

Dear Dr Racz

On behalf of the Editors, I am pleased to inform you that your Manuscript RSOS-191385 entitled "Usage frequency and lexical class determine the evolution of kinship terms in Indo-European" has been accepted for publication in Royal Society Open Science subject to minor revision in accordance with the referee suggestions. Please find the referees' comments at the end of this email.

The reviewers and handling editors have recommended publication, but also suggest some minor revisions to your manuscript. Therefore, I invite you to respond to the comments and revise your manuscript.

- Ethics statement

- Data accessibility

<http://datadryad.org/submit?journalID=RSOS&manu=RSOS-191385>

- Competing interests

- Authors' contributions

- Acknowledgements

- Funding statement

Please ensure you have prepared your revision in accordance with the guidance at <https://royalsociety.org/journals/authors/author-guidelines/> -- please note that we cannot publish your manuscript without the end statements. We have included a screenshot example of

the end statements for reference. If you feel that a given heading is not relevant to your paper, please nevertheless include the heading and explicitly state that it is not relevant to your work.

Because the schedule for publication is very tight, it is a condition of publication that you submit the revised version of your manuscript before 17-Sep-2019. Please note that the revision deadline will expire at 00.00am on this date. If you do not think you will be able to meet this date please let me know immediately.

Please note that Royal Society Open Science charge article processing charges for all new submissions that are accepted for publication. Charges will also apply to papers transferred to Royal Society Open Science from other Royal Society Publishing journals, as well as papers

submitted as part of our collaboration with the Royal Society of Chemistry (<http://rsos.royalsocietypublishing.org/chemistry>).

If your manuscript is newly submitted and subsequently accepted for publication, you will be asked to pay the article processing charge, unless you request a waiver and this is approved by Royal Society Publishing. You can find out more about the charges at <http://rsos.royalsocietypublishing.org/page/charges>. Should you have any queries, please contact opencscience@royalsociety.org.

Kind regards,
Andrew Dunn
Royal Society Open Science Editorial Office
Royal Society Open Science
opencscience@royalsociety.org

on behalf of Dr Alecia Carter (Associate Editor) and Kevin Padian (Subject Editor)
opencscience@royalsociety.org

Associate Editor Comments to Author (Dr Alecia Carter):

Associate Editor: 1

Comments to the Author:

Dear authors,

I have now received two reviews of your manuscript. Both reviewers are complimentary of the study and provide some helpful feedback. Please take care when addressing the reviewers' comments, particularly those regarding the clarity of the highlighted sections.

Thank you for submitting your manuscript to RSOS.

Reviewer comments to Author:

Reviewer: 1

Comments to the Author(s)

This paper concerns the evolution of kinship words in Indo-European languages. The two main questions asked are:

- (1) is the rate of replacement of kinship words in IE languages mediated by frequency of use of these words?
- (2) if so, is the rate comparable with other core vocabulary or similar to it?

The findings suggest that (perhaps unsurprisingly) central kinship words behave similarly to other core vocabulary words, in that rates of replacement are higher for infrequent words and lower for frequent words. The most newsworthy finding is - to my mind - the answer to the second question, namely that the effect is stronger for kinship words than for other core vocabulary words. In other words, if we believe the frequently-used-leads-to-slowly-replaced paradigm, then kinship words might be considered the more prototypical examples of such trends.

The only real quibble I had was with regard to the use of the term "closed lexical class". Generally speaking, "closed classes" refer to syntactic categories (nouns, verbs, adjective, etc.),

whereas this paper is dealing with a semantic category (i.e., kinship). I am personally not familiar with the terms “closed”/“open” being used in relation to semantic classes (as used here).

Further to this, I am not sure which “open class” the authors have in mind with regards to the comparison with kinship words – picked up in the discussion (this is the assertion on page 10-11, lines 275-278). I am not necessarily disagreeing with the assertion made there, but that first point in the discussion had me rather confused.

I was also wondering if the authors may have considered comparing the strength of the relationship between frequency of use and replacement words of kinship words with the relationship of frequency and replacement rates of number words. Number words are similarly a “system” semantic group – in that number words form a system together, in a way that kinship words may be said to form a system together. There is an older paper by Dehaene, S., & Mehler, J. (1992) looking at frequency of use of various number words in a bunch of IE languages which is interesting in this regard (see reference details at the end of this review).

As a general comment, I think the discussion could be developed further. For instance, this discussion of kinship words poses a few questions for future research: (1) in light of these results, should core vocabulary lists include more kinship terms, (2) are these results about kinship words specifically, or about groups of concepts which form such a “system” together (like kinship, numbers, maybe pronouns)? (3) does this trend uncovered for kinship words relate to kinship, or might it be a correlate of the fact that kinship terms are generally monosemous (rather than polysemous)?

Overall, the paper makes a worthy contribution to the development of kinship systems in IE languages and to language change in general and as such, I feel it should be published. The organisation and language of the paper are both of very high standard.

One small typo I noticed, page 12, line 334: a space is needed after the colon.

Reference:

Dehaene, S., & Mehler, J. (1992). Cross-linguistic regularities in the frequency of number words. *Cognition*, 43(1), 1-29.

Reviewer: 2

Comments to the Author(s)

The authors repeated a previous study on the correlation between word usage frequency and its rate of replacement along phylogeny on a closed lexical class. This is the first study looking at this problem on a closed lexical class, so the study definitely provides new insights into language evolution. I don't have the expertise to comment on the data. In terms of the statistical analyses, the study uses basically the same approach as the previous study, so the analyses seem to be valid. However, there is few description on the linear model. I have to go to the supps for details. Yet, even in the supps, the description is not clear, only two tables listing the fixed effect, without any explanation on what is control model 1 and control model 1b. In the main text, it says that the random effect structure was selected using model comparison. But there is no corresponding result and no description on the tested random effect structure. So I recommend the authors add more description on the linear model.

Author's Response to Decision Letter for (RSOS-191385.R0)

See Appendix A.

Decision letter (RSOS-191385.R1)

23-Sep-2019

Dear Dr Racz,

I am pleased to inform you that your manuscript entitled "Usage frequency and lexical class determine the evolution of kinship terms in Indo-European" is now accepted for publication in Royal Society Open Science.

on behalf of Dr Alecia Carter (Associate Editor) and Kevin Padian (Subject Editor)
openscience@royalsociety.org

Associate Editor Comments to Author (Dr Alecia Carter):
Associate Editor: 1
Comments to the Author:
(There are no comments.)

Reviewer comments to Author:

Appendix A

Dear Dr Carter

Please find attached the revised version of our ms ID RSOS-191385 submitted to Royal Society Open Science. We are very grateful to the editor and the reviewers for their comments and for their time. Our changes to the manuscript are in blue in the marked pdf file.

Please find below our detailed responses to reviewers, where responses are in italics.

Sincerely,
Péter Rácz, on behalf of
Sam Passmore
Catherine Sheard
Fiona Jordan

Reviewer 1

(...) The only real quibble I had was with regard to the use of the term “closed lexical class”. Generally speaking, “closed classes” refer to syntactic categories (nouns, verbs, adjective, etc.), whereas this paper is dealing with a semantic category (i.e., kinship). I am personally not familiar with the terms “closed”/“open” being used in relation to semantic classes (as used here).

Further to this, I am not sure which “open class” the authors have in mind with regards to the comparison with kinship words – picked up in the discussion (this is the assertion on page 10-11, lines 275-278). I am not necessarily disagreeing with the assertion made there, but that first point in the discussion had me rather confused.

We would argue that the closed / open distinction cuts across the function / content distinction for word classes. Function word classes, like English prepositions, can take up new members, such as "concerning" or "including". Content word classes, like Hungarian verbs, can be effectively closed in not taking up new members without a derivational suffix. This latter pattern is cross-linguistically common.

However, the reviewer is absolutely right that the classic use of open / closed is synonymous with content / function and that this is not clear in our text.

To clarify, we added the following paragraph (149):

"In most descriptions, the terminology of open/closed and content/function are used interchangeably for word classes. Here we step away from this and use 'closed semantic class' to refer to a set of closely related words where the head words can be listed exhaustively. In contrast, an open semantic class has loose (if any) connections between its members and can theoretically be of any size."

We then go on to define the core vocabulary as an "open semantic class" as per the above definition (l81) and refer back to it explicitly in the discussion (l195).

We hope this clarifies what we mean and, crucially, the basis of our comparisons.

I was also wondering if the authors may have considered comparing the strength of the relationship between frequency of use and replacement words of kinship words with the relationship of frequency and replacement rates of number words. Number words are similarly a "system" semantic group – in that number words form a system together, in a way that kinship words may be said to form a system together. There is an older paper by Dehaene, S., & Mehler, J. (1992) looking at frequency of use of various number words in a bunch of IE languages which is interesting in this regard (see reference details at the end of this review).

As a general comment, I think the discussion could be developed further. For instance, this discussion of kinship words poses a few questions for future research: (1) in light of these results, should core vocabulary lists include more kinship terms, (2) are these results about kinship words specifically, or about groups of concepts which form such a "system" together (like kinship, numbers, maybe pronouns)? (3) does this trend uncovered for kinship words relate to kinship, or might it be a correlate of the fact that kinship terms are generally monosemous (rather than polysemous)?

We thank the reviewer for this reference on the frequency of use of numerals which we now cite.

The reviewer's comments here point to an important issue that we did not unpack in sufficient detail in the discussion: the behaviour of kinship words vis-a-vis the core vocabulary both lends further credit to the long tradition of using the core vocabulary as a proxy of the entire vocabulary (afterall, we replicate a pattern with a different set of words) and provides a warning that the core vocabulary might not be enough, as the behaviour of kinship words is likely not idiosyncratic but rather indicative of closed semantic classes in general. We inserted a paragraph into the Discussion to consider this (l249):

"On a more general note, historical linguists can find comfort in our results which provide additional support to the use of the core vocabulary to establish a correlation between frequency of use and rate of replacement. However, these results also indicate that closed semantic classes behave differently and so across-the-board treatments of language change result in a loss of valuable resolution. The distinctness of kinship terms likely generalises to other closed classes. This invites further cross-linguistic research on word frequency and language change in other classes, such as colours or numerals (Berlin 1991, Dehaene 1992)."

We think that, for instance, the rate of replacement of numerals is a great topic for future research, but it would take a considerable amount of effort to integrate it into the current ms.

The reviewer makes an interesting point on the polysemy of kinship words; while the reviewer is probably right in that polysemy is relatively rare for this class, kinship-based metaphor is rampant and has not been extensively analysed using quantitative methods. This is, again, a valuable topic for future research but not something we would consider pursuing within the constraints of the present ms.

One small typo I noticed, page 12, line 334: a space is needed after the colon.

Corrected. Thanks!

Reviewer 2

(...) In terms of the statistical analyses, the study uses basically the same approach as the previous study, so the analyses seem to be valid. However, there is few description on the linear model. I have to go to the supps for details. Yet, even in the supps, the description is not clear, only two tables listing the fixed effect, without any explanation on what is control model 1 and control model 1b. In the main text, it says that the random effect structure was selected using model comparison. But there is no corresponding result and no description on the tested random effect structure. So I recommend the authors add more description on the linear model.

We made the following changes based on the reviewer's comments:

- *In the supplementary material, we now spell out the model formulae for all regression models.*
- *We justify the random effect structure: for the control models, there are practical grounds for not having random slopes. For the predictive model, we used goodness-of-fit tests; we now report the test statistics.*
- *We added more clarificatory text to make this section of the supplementary material easier to follow.*

We would argue that it is not necessary to expand the main text after these changes. This is because the main text does spell out the outcome and the predictors for the predictive model and does, concisely, set the basis of model selection. All of this is now discussed in more detail in the supplementary material.